# A Novel Approach to Determining Tumor Progression Using a Three-Site Pilot Clinical Trial of Spectroscopic MRI-Guided Radiation Dose Escalation in Glioblastoma

Karthik K. Ramesh [1,2], Vicki Huang [1,2], Jeffrey Rosenthal [3], Eric A. Mellon [4], Mohammed Goryawala [5], Peter B. Barker [6], Saumya S. Gurbani [1], Anuradha G. Trivedi [1,2], Alexander S. Giuffrida [1,2], Eduard Schreibmann [1], Hui Han [7], Macarena de le Fuente [8], Erin M. Dunbar [9], Matthias Holdhoff [10], Lawrence R. Kleinberg [11], Hui-Kuo G. Shu [1,12], Hyunsuk Shim [1,2,3,12,*] and Brent D. Weinberg [3,12,*]

1   Department of Radiation Oncology, Emory University School of Medicine, Atlanta, GA 30322, USA
2   Department of Biomedical Engineering, Georgia Institute of Technology, Atlanta, GA 30322, USA
3   Department of Radiology and Imaging Sciences, Emory University School of Medicine, Atlanta, GA 30322, USA
4   Department of Radiation Oncology, University of Miami, Miami, FL 45056, USA
5   Department of Radiology, University of Miami, Miami, FL 45056, USA
6   Department of Radiology and Radiological Science, Johns Hopkins University, Baltimore, MD 21218, USA
7   Biomedical Imaging Research Institute, Cedars-Sinai Medical Center, Los Angeles, CA 90048, USA
8   Department of Neurology, University of Miami, Miami, FL 45056, USA
9   Department of Neuro-Oncology and Neurosurgery, Piedmont Atlanta Hospital, Atlanta, GA 30309, USA
10  Department of Oncology, Johns Hopkins University, Baltimore, MD 21218, USA
11  Department of Radiation Oncology, Johns Hopkins University, Baltimore, MD 21218, USA
12  Winship Cancer Institute, Emory University School of Medicine, Atlanta, GA 30322, USA
*   Correspondence: hshim@emory.edu (H.S.); brent.d.weinberg@emory.edu (B.D.W.); Tel.: +1-(404)-778-4564 (H.S.)

**Abstract:** Glioblastoma (GBM) is a fatal disease, with poor prognosis exacerbated by difficulty in assessing tumor extent with imaging. Spectroscopic MRI (sMRI) is a non-contrast imaging technique measuring endogenous metabolite levels of the brain that can serve as biomarkers for tumor extension. We completed a three-site study to assess survival benefits of GBM patients when treated with escalated radiation dose guided by metabolic abnormalities in sMRI. Escalated radiation led to complex post-treatment imaging, requiring unique approaches to discern tumor progression from radiation-related treatment effect through our quantitative imaging platform. The purpose of this study is to determine true tumor recurrence timepoints for patients in our dose-escalation multisite study using novel methodology and to report on median progression-free survival (PFS). Follow-up imaging for all 30 trial patients were collected, lesion volumes segmented and graphed, and imaging uploaded to our platform for visual interpretation. Eighteen months post-enrollment, the median PFS was 16.6 months with a median time to follow-up of 20.3 months. With this new treatment paradigm, incidence rate of tumor recurrence one year from treatment is 30% compared to 60–70% failure under standard care. Based on the delayed tumor progression and improved survival, a randomized phase II trial is under development (EAF211).

**Keywords:** glioblastoma; GBM; spectroscopy; spectroscopic MRI; MRSI; radiation dose-escalation

## 1. Introduction

Glioblastoma (GBM) is a highly aggressive cancer that originates in glial cells in the brain, with a very poor prognosis for every age group. The disease has a median survival of 15–16 months [1,2], a median progression-free survival (PFS) of 4–8 months [2,3], and an annual incidence of 3.19 per 100,000 in the United States [4]. Very few interventions in recent decades have significantly improved outcomes for these patients. Standard therapies for GBM involve the surgical resection of visible tumor followed by radiation therapy (RT)

with concurrent and adjuvant temozolomide (TMZ) chemotherapy. Medical imaging plays a central role in this treatment pipeline with clinical MRIs guiding the surgical team and RT planning as well as the follow-up phase after initial treatment where patients are monitored for symptoms and imaging is assessed for concerning abnormalities.

Standard-of-care therapy for GBM patients is guided by a combination of T1-weighted contrast-enhanced MRI (T1w-CE) and T2-weighted fluid-attenuated inversion recovery MRI (FLAIR). T1w-CE indicates areas of high-grade tumor where gadolinium-based contrast agents can accumulate due to breakdown of the blood–brain barrier and is used to guide surgical resection. However, T1w-CE MRI usually underestimates the extent of tumor infiltration [5]. FLAIR identifies areas of surrounding tumor infiltration but cannot differentiate tumor from edema, inflammation, and radiation effects. In standard clinical practice, T1w-CE lesions combined with the resection cavity are targeted with a higher radiation dose (usually 60 Gy) and surrounding FLAIR hyperintensities are targeted with a lower radiation dose (45–54 Gy), both delivered over 30 fractions [6]. However, these techniques may not adequately target tumor, and imaging methods that detect tumor with high specificity while preserving normal brain may improve outcomes for GBM patients. We have been developing a high-resolution, 3D magnetic resonance spectroscopic imaging (MRSI) sequence with whole brain coverage that we have termed spectroscopic MRI (sMRI) [7]. sMRI is a non-invasive metabolic imaging technique that can non-invasively detect tumor without contrast agents by detecting areas of metabolically abnormal tumor which have elevation of choline (Cho) due to high cell turnover and decreased N-acetylaspartate (NAA) due to displacement of normal neurons and axons by tumor [8–10]. In addition, some studies have found a relationship between Cho levels and grade of astrocytoma, with a higher grade being associated with higher Cho [11]. The ratio of metabolite signals Cho to NAA (Cho/NAA) is highly associated with the presence of tumor even in regions of the brain that appear healthy on standard MRI sequences [8,12,13], and a Cho/NAA ratio greater than two times that in normal-appearing white matter (NAWM) are correlated with regions of tumor recurrence in 81% of patients [14]. This information can be used in brain tumor patients to treat invasive tumor with higher radiation doses while minimizing doses to normal brain, potentially delaying recurrence for GBM patients without excessive risk of toxicity.

We recently completed a multi-institutional, single-arm pilot clinical trial for newly-diagnosed GBM patients where brain regions with a Cho/NAA $\geq$ 2x normal were treated with an escalated radiation dose of 75 Gy [7]. Patients had a median overall survival (OS) of 23.0 months, seven months longer than that in historical controls [2,15–17]. During the post-treatment follow-up period, serial MRIs including T1w-CE and FLAIR were acquired at routine intervals, but the interpretation of post-treatment imaging was confounded by the overlapping appearance of tumor recurrence and post-treatment effects including pseudo-progression and radiation necrosis. Both pseudo-progression and radiation necrosis are phenomena characterized by worsening of T1w-CE enhancement and FLAIR hyperintensity after RT due to tissue injury from radiation, with pseudo-progression occurring within 90 days of radiation and radiation necrosis peaking 1 to 2 years post-RT [18,19]. We saw these effects exacerbated in our pilot clinical trial where larger volumes of tissue were treated with escalated radiation dosage. While sMRI could be helpful with tumor detection after treatment, extensive necrosis can lead to a low Cho peak, with signals from lipid and other metabolites such as lactate often suppressing the Cho signal [10,20]. Further, sMRI acquisitions were not available during follow-up for these patients.

To address these challenges, we developed novel quantitative and visualization tools to track patients after initial treatment, minimize the effects of confounding treatment changes, pinpoint tumor recurrence timepoints, and perform PFS analysis. These tools use a combination of a quantitative imaging platform and structured reporting methods. The Brain Imaging Collaboration Suite Longitudinal Imaging Tracker (BrICS-LIT) is a custom-built web application for the purpose of longitudinal MRI imaging follow-up of brain tumor patients [21,22]. Within BrICS-LIT, lesions on FLAIR and T1w-CE imaging are

segmented and displayed, radiation dose maps can be overlaid upon anatomic imaging, and patient clinical information can be displayed. Structured reporting can be used for a more objective interpretation of the patient's disease status and has gained popularity for applications such as breast and prostate imaging [23,24], but remains underutilized in brain tumor imaging. RANO, or the Response Assessment in Neuro-Oncology criteria, has found success in clinical trial applications but is less commonly used in general clinical practice. While we previously reported the (PFS) from our clinical trial, we did not provide details as to how we determined tumor progression, especially when considering the high rates of pseudo-progression and radiation necrosis [7]. In this study, we demonstrate how we determined progression for these patients by applying the Brain Tumor Reporting and Data System (BT-RADS) framework of structured imaging classification to imaging obtained as part of our clinical trial [25–27]. BT-RADS scores and lesion volumes were calculated for the quantitative assessment of patient disease-status, and along with the visualization features, served to determine true tumor recurrence timepoints.

## 2. Materials and Methods

### 2.1. Clinical Trial Treatment Protocol

After newly diagnosed GBM patients completed surgery, sMRI scans were acquired along with standard imaging to guide radiation treatment. At each participating site, a Siemens 3T scanner was used to perform an echo planar sMRI pulse sequence with GRAPPA parallelization with either a 32- or 20-channel head coil. The scan was acquired with an echo time of 50 ms, repetition time of 1551 ms, and flip angle of 71 degrees, leading to a scan time of 15 min and a nominal voxel size of 108 μL [14,21]. Raw sMRI DICOM data were processed by the MIDAS software suite (University of Miami, Miami, FL, USA) [28], before being imported into the Brain Imaging Collaboration Suite (BrICS) for visualization and RT planning. Within BrICS, metabolite maps generated by MIDAS were registered and interpolated to the higher resolution T1w MRI space before being overlaid as heat maps. Contours containing Cho/NAA $\geq$ 2x normal were generated within BrICS for use as the high-dose radiation target and inspected by radiation oncologists at each respective site to ensure that these contours were appropriate and did not include organs-at-risk such as the brain stem to a significant extent.

In Figure 1, we show three example patients on the trial. For each patient, the contour which includes Cho/NAA $\geq$ 2x normal is significantly larger than the extent of contrast enhancement. In standard therapies, only the volume of contrast enhancement (indicated on the right for each patient) would have received high-dose radiation, potentially ignoring significant amounts of infiltrative tumor demarcated by the Cho/NAA metabolite maps. In our trial, the volumes on the left, represented by the 2x contours, received an escalated radiation dose of 75 Gy to potentially delay in-field tumor recurrence. Two additional radiation targets were generated for each patient on the trial—the resection cavity and any residual contrast within the cavity were prescribed 60 Gy and regions of FLAIR hyperintensity were prescribed 50.1 Gy of radiation. The latter two RT targets had a 5 mm margin expansion to account for microscopic invasion of tumor, and all three targets had an additional 3 mm margin added to generate planning target volumes. Radiation was split into 30 fractions and administered over a six-week period. Concurrent TMZ chemotherapy was also administered at standard-of-care dosing (75 mg/m$^2$/day every day) during RT. After completion of RT, adjuvant TMZ was administered for each patient (150–200 mg/m$^2$/day days 1–5 every 28 days) per standard-of-care [29].

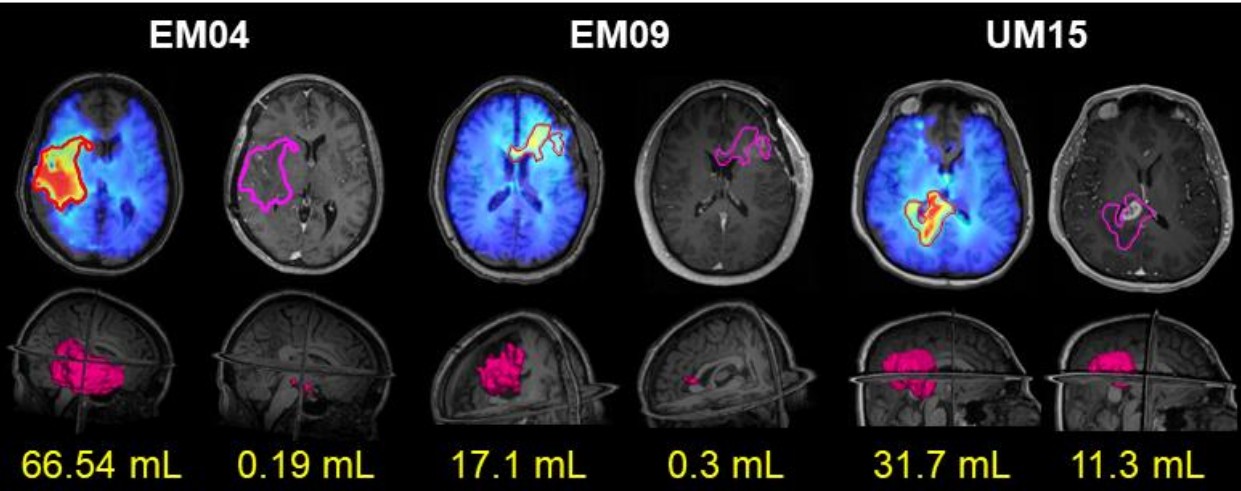

**Figure 1.** A comparison between the ratio map of choline to N-acetylaspartate (Cho/NAA) and T1-weighted contrast-enhanced (T1w-CE) imaging for three patients on the trial. For each patient, the top left image contains the Cho/NAA ratio map superimposed on the T1w image along with a contour encompassing all voxels with a Cho/NAA $\geq 2x$ normal. That contour is overlaid on the T1w-CE image on the top right image which shows enhancement from residual tumor and recent surgery. The bottom left image shows a 3D volume rendering of the volume of tissue that received 75 Gy of radiation guided by the Cho/NAA $\geq 2x$ contour, while the bottom right image shows a 3D volume rendering of the residual contrast enhancing lesion, which normally receives 60 Gy of radiation along with resection cavity in standard therapies. In all cases, the Cho/NAA $\geq 2x$ contour was significantly larger than the contrast-enhancing lesion.

For the cohort of 30 patients who received treatment, 2 had an IDH mutation, 9 were MGMT hypermethylated, 11 had a gross total resection (defined by whether their pre-RT T1w-CE lesion volumes were less than 1 mL), and 19 had a subtotal surgical resection [7]. Informed consent was acquired for every patient in this trial prior to their first sMRI scan, with acknowledgment that follow-up imaging would be collected to monitor treatment response. Both the study and informed consent were approved by the Institutional Review Boards (IRBs) at each participating institution. The following analyses were performed with de-identified datasets from the trial.

### 2.2. Follow-Up

After patients completed RT, they returned for follow-up imaging every 2–3 months. At University of Miami and Emory University, follow-up occurred every 2 months, while patients at Johns Hopkins University had follow-up visits at 3-month intervals. Standard clinical follow-up was performed using the institution standard brain tumor protocol. T2 FLAIR and T1w-CE MRIs from each follow-up visit were uploaded to BrICS-LIT [22]. In BrICS-LIT, each MRI uploaded is rigid registered and interpolated to a higher-resolution, thin-sliced atlas MRI to enable slice-by-slice comparisons across extended time. Enhanced lesions on T1w-CE MRIs and hyperintensity on FLAIR MRIs were contoured in a semi-automated manner with manual editing and inspection by clinicians afterwards [22]. Clinical information about the patient such as the medication they were prescribed, their genetics, and treatment-related information were stored in a REDCap database. BrICS-LIT is configured to automatically retrieve relevant, de-identified clinical information from REDCap. With these segmented lesion volumes, as well as relevant clinical information about the patient, our neuro-radiologist prospectively assigned BT-RADS scores for each follow-up date. In cases with indeterminate imaging findings, additional clinical imaging sequences, such as diffusion-weighted imaging (DWI), and dynamic susceptibility perfusion-weighted imaging (PWI) were reviewed.

In Figure 2, we show the follow-up imaging of a patient on the trial in BrICS-LIT. Users can simultaneously view 5 study dates at a time and scroll through previous follow-up dates that are available, enabling a comprehensive view of disease state over time. Within BrICS-LIT, segmented contours are highlighted in red, with lesion volumes and BT-RADS scores for each date displayed below. As an example, the earliest follow-up date on the right received a baseline BT-RADS score of 0 as it was immediately post-surgery and pre-RT. The four follow-up dates to the left occurred 1–11 months post-RT. The next follow-up date to the left received a score of 3A due to increased lesion volumes suggesting a possible cause for concern, however, as the imaging was performed less than 3 months post-RT it more likely reflected pseudo-progression rather than true tumor progression. The imaging in the subsequent two dates received scores of 1A and 2, indicating improvement and stable control of disease, as the lesion volumes remained relatively stable compared to previous follow-ups. The far-left study date, occurring 11 months after completion of RT, received a score of 4 due to a large increase in FLAIR and T1w-CE lesion volumes. In the bottom row of the figure, a radiation dose map is overlaid on T1w-CE imaging, which can help determine whether a patient has experienced out-of-field recurrence, common signs of true tumor progression. This patient's enhancing lesion in T1w-CE started in-field and spread slightly outside of the high-dose boundary margins delineated in red, suggesting possible marginal failure.

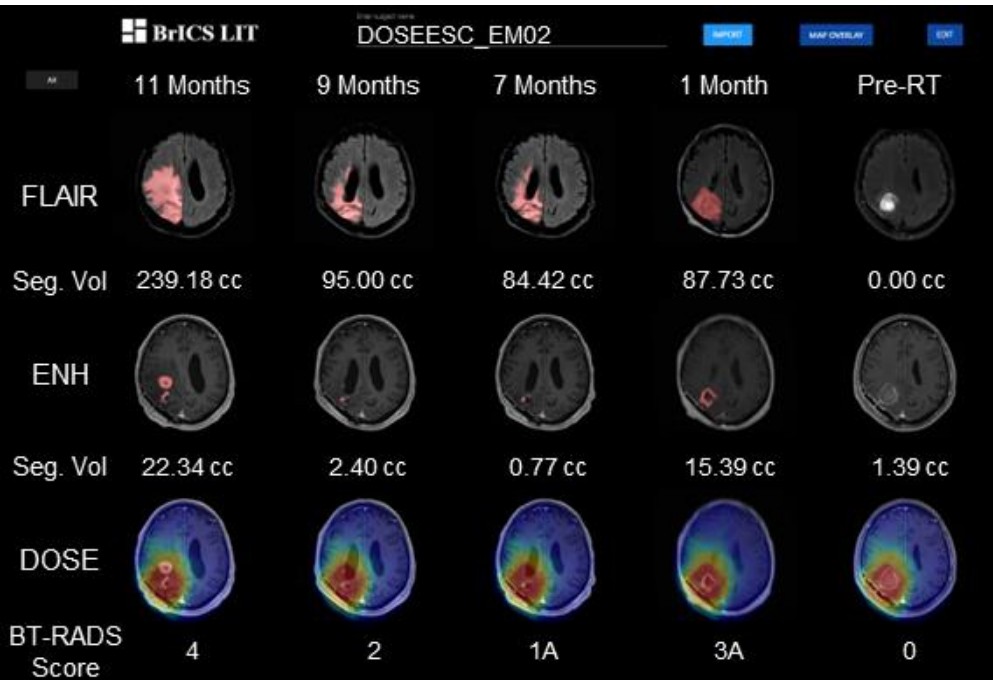

**Figure 2.** A view of the Brain Imaging Collaboration Suite Longitudinal Imaging Tracker (BrICS-LIT). The top row consists of T2-weighted fluid-attenuated inversion recovery (FLAIR) MRIs and the middle row of T1w-CE MRIs co-registered and interpolated to an atlas for easier comparison. Studies are longitudinally arranged from newest (left column) to oldest (right column). Segmented lesions are overlaid in the top two rows, while the radiation dose map is overlaid on the T1w-CE MRI in the bottom row to intuitively determine the radiation dose received of the suspected recurring tumor.

Often, if an enhancing lesion grows suspiciously fast, patients will undergo additional surgeries during this follow-up period. Many patients on this trial had second or even third surgeries to remove an enhancing lesion that was suspicious for tumor. After those surgeries, clinical pathologists at each respective site assessed tissue samples for presence of tumor and reported findings in pathology reports.

*2.3. Tumor Recurrence Determination*

With BrICS-LIT, we graphed lesion volumes and BT-RADS scores over each patient's follow-up period to observe long-term trends. As tumor-mimicking treatment effects such as pseudo-progression and radiation necrosis are commonly associated with high-dose radiation, we sought to use BrICS-LIT and clinical pathology reports to determine true tumor recurrence dates on a case-by-case basis. The following criteria were established across all patients to label the date of first disease progression. First, if the patient received any follow-up surgery and biopsy, then their pathology report served as the ground truth for tumor progression. If the re-resection or biopsy specimens contained over 20% tumor as described in the pathology report, the previous date was labeled as progression. When the pathology report did not include a percentage but mentioned positive tumor presence, the previous follow-up was labeled as progression. While other factors included in pathology reports such as mitosis or elevated proliferation index could have been used as cutoff points for progression, these factors were not present in every report we received. Therefore, we used a defined cutoff based on information that every report shared. This cutoff was set a priori and not changed in the analysis. For patients who did not receive re-resection, follow-up imaging was evaluated by comparing volumetric changes of contrast-enhancing and FLAIR lesions over time. Dates where significant increases in segmented tumor volume leading to BT-RADS scores of 3c and 4 were further examined using RT dose map overlays. By overlaying an RT dose map over follow-up anatomic imaging, we could look for cases where an enhanced lesion continued to expand and significantly spread outside of the high-dose target (appearing red in the overlaid dose maps). If the enhancing lesion spread outside the high-dose radiation field, the corresponding follow-up date was defined as the tumor recurrence date. Otherwise, if the enhancing lesion volume stabilized within the high-dose region of the brain, the enhancement was attributed to radiation necrosis. Every case where the enhancing lesion extended past the high-dose radiation target was verified and agreed upon by a neuro-radiologist and the patient's treating radiation oncologist.

*2.4. Survival Analysis*

All survival outcomes were assessed using the lifelines Python survival analysis library [30]. PFS curves were calculated starting from the date of surgery using the Kaplan–Meier estimator method [31]. The PFS was defined from the date of surgery to first disease recurrence. If a patient died from known causes or surgeries before confirmed tumor recurrence on imaging, their date of death was used as the progression date. For these cases, often the last follow-up imaging acquired was very close to their date of survival. For patients who had not yet progressed or who were lost to follow-up, their last date of contact without tumor recurrence was right-censored.

## 3. Results

*3.1. Survival Analysis*

Eighteen months after completing enrollment, the median PFS time was 16.6 months (Figure 3) with a median time to follow-up of 20.3 months in censored patients. The Kaplan–Meier estimator with 95% confidence intervals is included. Out of the 30 patients enrolled and who completed treatment, 19 patients had confirmed disease progression at the time of last follow-up. Of the 19 patients either with disease progression or who expired before evidence of progression, 7 had tumor recur within the high-dose radiation field, 3 patients had out-of-field recurrence, 4 patients had new lesions discovered in areas that received minimal to zero radiation, and 5 patients died prior to imaging progression. Out of the 11 censored patients, 5 are still alive without signs of disease progression in follow-up imaging, and 6 are dead without sufficient follow-up imaging to determine when or if tumor recurred before their deaths. These patients were often lost to follow-up with their death date independently confirmed via obituary. For these patients that did die, their last date of contact without tumor recurrence was used and right-censored because of lack of information about the cause of death.

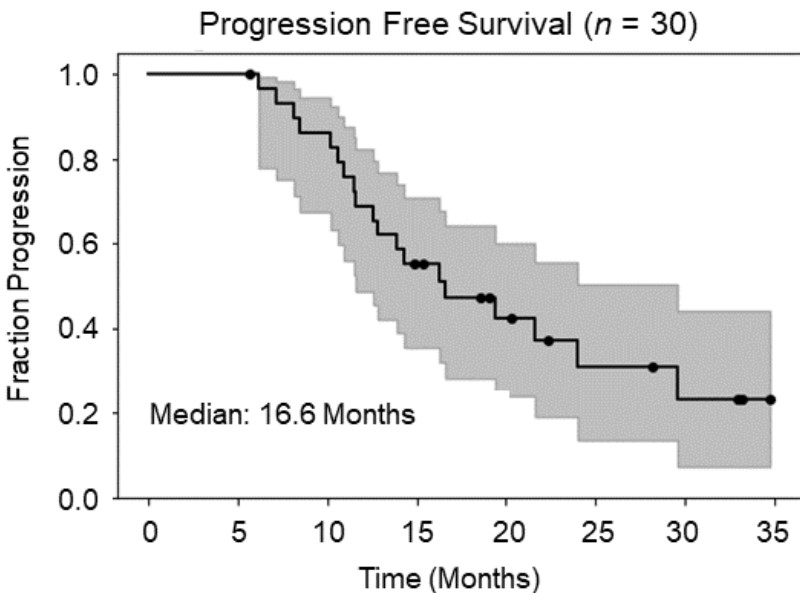

**Figure 3.** Kaplan–Meier estimator for PFS; median PFS: 16.6 months with a median time to follow-up of 20.3 months.

As mentioned previously, fourteen out of nineteen patients with tumor progression had confirmed progression before their deaths. The criteria we outlined for determining progression were applied to this subgroup with additional details in Table 1. For this patient subgroup, one had an IDH mutation, while five were MGMT hypermethylated status. Due to a small sample size, it is difficult to discern any relationship between, for example, MGMT status and progression type or PFS. However, three of the five patients with MGMT hypermethylation had a PFS that falls below the median for the cohort; however, this could also be due to the post-op residual enhancing volume, which we label as the extent of resection (EOR) and other factors. Of note, patients with multifocal lesions, which received minimal to zero radiation or surgical intervention, led to poorer outcomes and lower PFS values.

**Table 1.** Compiled genetic and PFS data for 14 of the 19 patients with confirmed tumor progression.

| Patient | PFS (Months) | IDH | MGMT | EOR (cc) | Progression Type |
|---|---|---|---|---|---|
| 1 | 19.0 | 1 | 1 | 1.6 | In-Field |
| 2 | 14.3 | 0 | 0 | 2.6 | In-Field |
| 3 | 16.6 | 0 | 0 | 0.3 | Out-of-Field |
| 4 | 12.6 | 0 | 1 | 2.2 | In-Field |
| 5 | 13.9 | 0 | 0 | 0.5 | In-Field |
| 6 | 8.4 | 0 | 0 | 1.4 | In-Field |
| 7 | 8.1 | 0 | 0 | 12.1 | Multifocal |
| 8 | 6.2 | 0 | 0 | 3.9 | Multifocal |
| 9 | 10.2 | 0 | 1 | 7 | Out-of-Field |
| 10 | 7.1 | 0 | 0 | 0.4 | Multifocal |
| 11 | 11.0 | 0 | 1 | 5 | Multifocal |
| 12 | 11.5 | 0 | 0 | 2 | In-Field |
| 13 | 16.2 | 0 | 0 | 0.5 | In-Field |
| 14 | 29.6 | 0 | 1 | 2.9 | Out-of-Field |

*3.2. Example Assessments of Recurrence Using Patient Follow-Up*

Due to the difficulty of distinguishing tumor progression from necrosis in a systematic manner when follow-up surgical pathology was not available, we assessed patient follow-up data on a case-by-case basis. Representative examples of imaging evaluation are included below.

The first case is an IDH wild-type, MGMT hypermethylated patient who exhibited progression-like imaging characteristics that were ultimately attributed to treatment effects. Figure 4B graphs this patient's BT-RADS scores from their initial surgery before RT until their latest follow-up visit. This patient had two surgical resections after RT marked with red stars. After the first surgery, the neuropathologist found minimal tumor even after a one-year gap from completion of RT. About nine months later, the patient received a second surgical resection due to increased enhancement volume on clinical imaging; 80% of sampled cells were necrotic. Leading up to each follow-up surgery, FLAIR and T1w-CE lesion volumes trended upward leading to BT-RADS scores generally above 2, suggesting the possible presence of tumor. Figure 4C shows follow-up imaging that corresponds to each visit in the graph above. While FLAIR lesion volumes appear to increase after month 12 as indicated by Figure 4A, the tumor does not appear to spread contralaterally or compress against tissue in the midline of the brain as is visible in the top row of Figure 4C. The middle row of Figure 4C shows enhancing volumes from T1w-CE MRIs. The rim of enhancement around the surgical resection cavity appears to spread radially. Despite this, the enhanced rim stays within the high-dose radiation field demarcated in red in the third row. Despite trends in increasing enhancing lesion in Figure 4A, the volume stabilizes after month 20. During standard clinical follow-up, the increasing enhanced rim and FLAIR hyperintense volumes would lead to an early designation of tumor recurrence. However, both pathology reports in this case failed to identify significant amounts of tumor, suggesting that most enhancement and FLAIR lesions were due to inflammation and necrosis from high-dose radiation for years after they completed RT. This patient is still alive.

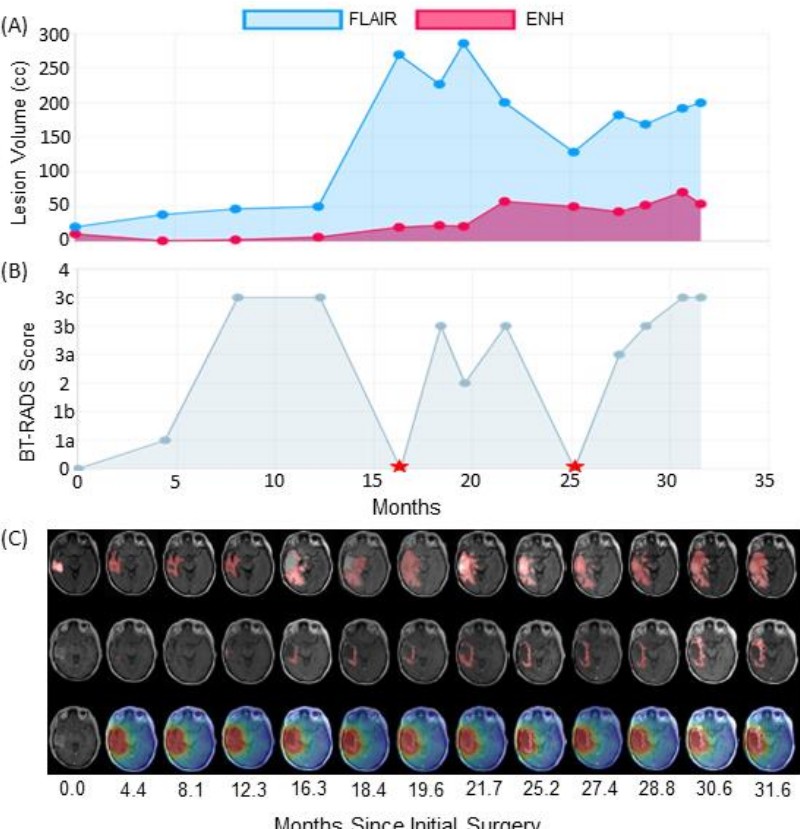

**Figure 4.** With our web platform, clinicians assessed (**A**) graphs of lesion volumes from FLAIR and T1w-CE (labeled ENH) imaging, as well as (**B**) A graph of Brain Tumor Reporting and Data System (BT-RADS) scores during a patient's follow-up. The red stars indicate re-resection of suspected lesions. In this case, the patient had two re-resections (16.3 and 25.2 months) with significant treatment effect and necrosis found in both cases. (**C**) For each follow-up visit in the above graph, the FLAIR MRIs (top row),

T1w-CE MRIs (middle row), and radiation dose map overlaid on T1w-CE MRIs (bottom row) are displayed from oldest (left column) to newest (right column). While the increasing enhancing rim as well as FLAIR volumes were cause for concern, this patient had still not had confirmed disease progression.

Figure 5 is an example patient with initial in-field tumor recurrence followed by out-of-field progression. In this case, the surgical pathology report indicated 30% tumor in sampled tissue, with the follow-up visit before re-resection (indicated by the red star) labeled as the tumor progression date (indicated by the orange diamond). This is a case that demonstrates the value of overlaying radiation dose maps. Even after this patient received a follow-up surgery, another enhancing lesion appeared on the contralateral side of the brain. This lesion is outside of the radiation treatment zone. Less than one month after this patient's follow-up surgery, their tumor had infiltrated contralaterally and was not evident in contrast enhanced imaging before their follow-up surgery. This second focal lesion continues to increase on both FLAIR and T1w-CE imaging shortly after. Both follow-up visits were assigned BT-RADS scores of 4, suggesting true tumor progression. By co-registering follow-up imaging and overlaying radiation dose maps, BrICS-LIT proves useful not only in retrospectively assessing imaging for tumor recurrence but also prospectively identifying out-of-field recurrence in this case. While FLAIR abnormality does spread contralaterally before the second surgery, this could have easily been attributed to treatment-related inflammation, especially due to lack of significant contrast enhancement in the same area.

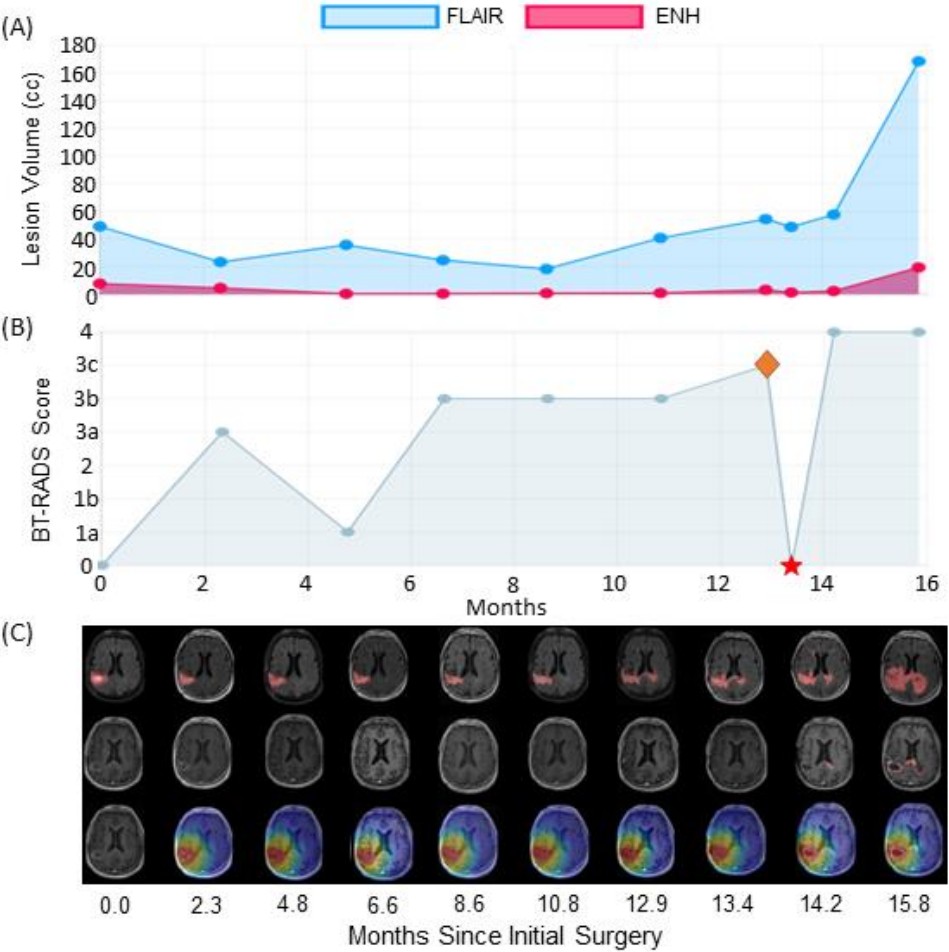

**Figure 5.** (**A**) A graph of changing lesion volumes from FLAIR and T1w-CE (labeled ENH) imaging can help assess suspicious trends, especially after re-resection in this case. (**B**) A graph of BT-RADS scores during a patient's follow-up. The red star indicates re-resection of suspected lesions. This patient had

one follow-up surgery that found 30% tumor in samples biopsied (13.4 months). Therefore, their recurrence date (orange diamond) was set before their follow-up surgery. (**C**) For each follow-up visit in the above graph, the FLAIR MRIs (top row), T1w-CE MRIs (middle row), and radiation dose map overlaid on T1w-CE MRIs (bottom row) are displayed. From the 14th month onwards, a second lesion is apparent on the contralateral side of the brain outside of the radiation treatment zone.

## 4. Discussion

In this study we describe the use of novel imaging visualization tools and structured reporting to analyze the results to date of our 30-patient clinical trial utilizing dose-escalated radiation guided by Cho/NAA sMRI maps. With standard-of-care therapy including standard RT with concurrent and adjuvant TMZ, GBM prognosis remains poor with a median OS that is consistently limited to about 16 months and an incidence of recurrence at one year from start of treatment of 60–70% [2]. Our results from our single arm study show a substantial improvement with a one-year incidence of recurrence of 30% and median PFS of 16.6 months. For the sake of comparison, Avastin, an anti-angiogenic therapy commonly used for newly-diagnosed GBM patients, minimally improved OS to a timepoint lower than the median PFS calculated in our pilot study (16.1 months) [15]. While these results are promising, we must caution that our pilot study had a much smaller sample size compared to historical controls and the Avastin trial and at the very least, justifies a larger consortium-level trial to determine a true effect size. We have previously reported a median OS for this cohort of 23.0 months [7,32]. In another trial (NRG Oncology BN001 NCT02179086) that escalated radiation dose guided only by standard imaging [33], there was no significant difference in median OS between standard treatment groups and the dose-escalated group. Guiding radiation targets using sMRI led to larger, more specific treatment volumes that contain high quantities of tumor not visible in standard imaging. By using an imaging technique with a higher degree of specificity, we appear to show a lower one-year incidence of recurrence and improve survival outcomes in this cohort, justifying a larger prospective trial to assess the efficacy of sMRI-guided high-dose radiation.

While escalated radiation shows promising results, most patients on the trial exhibited pseudo-progression and radiation necrosis that lasted far longer than patients on standard radiation therapies. In standard clinical practice, T1w-CE imaging is the mainstay for defining tumor recurrence. However, under high-dose irradiation, post-treatment radiation effects often mimic those of tumor recurrence on clinical images, prompting labeling of disease progression before true tumor recurrence. Multiple patients on the trial were recommended for additional surgeries due to the apparent increase in contrast enhancing volume on T1w-CE imaging; however, these lesions were often in-field of the high-dose target and were later confirmed to be either pseudo-progression or radiation necrosis via pathology reports. The pathology reports for the first patient from our results section showed zero and minimal tumor both one year and two years after completion of RT. It was common to see pathology reports five to even nine months after completion of RT finding minimal tumor in samples biopsied. Our dependence on pathology reports from biopsy for in-field recurrence determination is a limitation for our criteria, especially in cases where there were not additional surgical interventions at critical timepoints.

While we acknowledge that quantitative tools such as BrICS-LIT do not necessarily solve the problem of differentiating tumor from necrosis, they were helpful in tracking the patient's disease-state during follow-up. Using our LIT database, radiation dose maps, and post-surgery pathology reports, we performed a retrospective analysis showing a ten month improvement of PFS compared to standard therapy using sMRI-guided dose escalation. With BrICS-LIT, we can graph lesion volumes and structured reporting scores throughout the patient's follow-up period. Rather than manually inspecting independent images and qualitatively assessing the patient across their latest few follow-up dates, with BrICS-LIT we can use segmentation algorithms to calculate lesion volumes and make comparisons quantitatively. If, for example, enhancing lesion volume is increasing at an exponential rate, there is good reason to be suspicious for tumor progression. However, if the lesion

volume stays within-field and remains stable over several months, the enhancing lesion could be mostly radiation necrosis unless proven otherwise through biopsy. A limitation of this study is that we did not use MRI perfusion-weighted imaging or other advanced imaging to further differentiate imaging findings. While some studies have shown benefits in using MRI perfusion to characterize indeterminate cases, there is wide variation in how it is applied, including at the three trial sites on this study, which would lead to variations in interpretation [34]. A further study is warranted to determine how to incorporate perfusion into follow-up imaging in the clinical trial setting. By using BrICS-LIT to overlay radiation dose maps, we can visualize whether the lesion has spread outside of the high-dose radiation target and find multi-focal lesions far more easily than the standard clinical tools available.

Future plans involve fully automating lesion segmentation in both FLAIR and T1w-CE MRIs as well as generating suggestive, structured reporting scores for each follow-up visit. While escalating the radiation dosage to 75 Gy led to minimal clinical toxicity for patients on this trial, there was one patient who had grade 3 toxicities as a result of their radiation, with edema on FLAIR imaging pushing across the midline of their brain. Additionally, there were a handful of cases where 75 Gy was not sufficient to control tumor as at least seven cases had in-field recurrence. External factors such as MGMT hypermethylation and IDH mutation likely contribute to how patients respond to different amounts of radiation [35]. Through the imaging data, radiation dosages, sMRI data, genetics, and medication data we have collected in BrICS-LIT as well as REDCap, we are currently working on a time-to-recurrence predictive algorithm that attempts to predict tumor recurrence risk for each voxel in pre-RT imaging. By generating Cho/NAA $\geq$ 2x, 3x, and 4x normal contours and calculating the median recurrence risk within each of these contours, we hope to generate automated radiation-dose plans within each Cho/NAA contour that reduces recurrence risk as much as possible while also minimizing patient toxicity. Finally, with the encouraging survival data from our pilot study, a randomized, phase II clinical trial (EAF211) is planned with the Eastern Cooperative Oncology Group (ECOG)-American College of Radiology Imaging Network (ACRIN) to guide escalated radiation with Cho/NAA sMRI maps.

## 5. Conclusions

With a median time to follow-up of 20.3 months for censored patients, guiding escalated radiation dose to glioblastoma patients through spectroscopic MRI led to a median PFS of 16.6 months with a one-year incidence of recurrence from treatment of 30%, half that of standard therapy. These results are extremely encouraging and suggest the need for sMRI to be a part of standard clinical practice. While escalated radiation doses over larger treatment volumes show promising results towards improving survival, differentiating between true tumor recurrence and radiation-related necrosis even one year after completion of radiation therapy remains a challenge.

By developing quantitative web applications such as BrICS-LIT, we have managed to segment lesions and calculate structured reporting scores for each patient during the follow-up phase of their treatment with the hope of more easily delineating recurrence. Through radiation map overlays, graphing lesion volumes for observable trends, and patient pathology reports, we present our methodology for calculating the tumor progression date. Through the quantitative data we have collected in BrICS-LIT, we plan to develop automated, optimized radiation treatment plans that operate at a voxel-level granularity, with the hope of delaying recurrence for as long as possible.

**Author Contributions:** Conceptualization, E.A.M., M.G., P.B.B., S.S.G., E.S., H.H., M.H., L.R.K., H.-K.G.S., H.S. and B.D.W.; Methodology, K.K.R., V.H., P.B.B., S.S.G., E.S., H.-K.G.S., H.S. and B.D.W.; Software, K.K.R., S.S.G., E.S., H.S. and B.D.W.; Validation, E.A.M., M.d.l.F., M.H., H.-K.G.S., H.S. and B.D.W.; Formal analysis, K.K.R., V.H., J.R., E.A.M., M.d.l.F., H.S. and B.D.W.; Resources, B.D.W.; Data curation, K.K.R., V.H., J.R., E.A.M., M.G., P.B.B., A.S.G., E.M.D., M.H., L.R.K., H.-K.G.S., H.S. and B.D.W.; Writing—original draft, K.K.R. and H.S.; Writing—review & editing, K.K.R., V.H., J.R., E.A.M., M.G., P.B.B., S.S.G., A.G.T., A.S.G., E.S., H.H., M.d.l.F., E.M.D., M.H., L.R.K., H.-K.G.S. and

B.D.W.; Visualization, K.K.R., A.G.T. and B.D.W. All authors have read and agreed to the published version of the manuscript.

**Funding:** This clinical trial was originally funded by NCI R01 CA214557 (HS, LK, HGS). The project described was supported by NIBIB U01 CA264039 (HS), NIH R01 NS121544 (HH), RSNA Research & Education Foundation through a Research Scholar Grant (BW). This work is also supported by pre-doctoral fellowship F31CA247564 (KR).

**Institutional Review Board Statement:** The study was conducted according to the guidelines of the Declaration of Helsinki and approved by the Institutional Review Board (or Ethics Committee) of Emory University (IRB00094188 on 4/14/2017).

**Informed Consent Statement:** Informed consent was obtained from all subjects involved in the study.

**Data Availability Statement:** The data presented in this study are available on request from the corresponding author.

**Acknowledgments:** The authors would like to acknowledge the amazing work and dedication by our multisite coordinator Latrisha Moore, our Institutional Review Board staff, as well as clinical research coordinators at each of our three sites who helped with collecting and de-identifying follow-up data and without whom, our efforts would have been immeasurably difficult. We would like to also thank all our patients and their caregivers who believed in our study.

**Conflicts of Interest:** The authors declare no conflict of interest.

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
