# Peer review of "A Novel Approach to Determining Tumor Progression Using a Three-Site Pilot Clinical Trial of Spectroscopic MRI-Guided Radiation Dose Escalation in Glioblastoma"

_tomography, doi:10.3390/tomography9010029_

Round 1

Reviewer 1 Report

I wish to congratulate the authors on presenting this data on a unique protocol and I look forward to further results and studies. This manuscript would be further improved by potentially including PFS data based on extent of resection, MGMT methylation status, and IDH status. MGMT methylated patients tend to have a higher chance of treatment effect which would be helpful to know for this cohort. 

Additionally, further clarification on using 20% as cut off for tumor progression on patients who underwent re-resection would be be beneficial. Were other parameters such as mitosis and elevated proliferation index also considered?

Lastly, were other advanced imaging techniques such as perfusion and/or spectroscopy considered or utilize? These are common protocols done at many brain tumor centers which also help with determination of treatment effect vs tumor progression. 

Reviewer 2 Report

The authors present a paper of clinical interest, which is illustrated by interesting images. The subject falls within the scope of the journal. Discussion of the findings are well done and well-founded. The bibliography is pertinent and current. However, the text needs improvement.

Did you use 1H Spectroscopic?

Why you did not evaluate perfusion to detect the tumoral invasion:?

Why did you use echo time of 50 ms? To detect choline compounds 144 ms would not be more adequate?

Some studies have suggested an association between tumor grade and Cho levels in atrocytomas, with higher-grade tumors having greater Cho concentrations. The latter finding may be absent in highgrade gliomas with extensive necrosis that tends to result a low choline peak. In this case, enhanced lactate and lipid concentrations usually suppress the peaks of the other metabolites, including Cho. Please, mention this topic in your paper and the following references (see file).

Fulham MJ et al. Mapping of brain tumor metabolites with protonMR spectroscopic imaging: clinical relevance. Radiology

Farche MK, et al. Revisiting the use of proton magnetic resonance spectroscopy in distinguishing between primary and secondary malignant tumors of the central nervous system. Neuroradiol J. 2022 Oct;35(5):619-626. 

The Cho/Cr ratio has been associated with the proliferation index (Ki-67), tumor density, and the degree of differentiation but is not specific.

The limitation is the small number of subjects (30).

Reviewer 3 Report

The authors analyzed the post-treatment imagistic and pathology data from a single-arm 3-side clinical trial which was done to assess the survival benefits of newly-diagnosed GBM patients treated with dose-escalated spectroscopic MRI (sMRI)-informed radiotherapy. The generation of sMRI data to better define the radiation field and dosages after surgery was deemed a success, with a gain of about seven months in median overall survival for these patients over historical controls. As expected, the radiation dose escalation in these patients also led to an increase in the size and duration of radiation adverse events (e.g., pseudo-progression and radiation necrosis), which complicates the process of interpretation of post-treatment imaging. To address these challenges, the authors developed novel quantitative and visualization tools based on the Brain Imaging Collaboration Suite Longitudinal Imaging Tracker (BrICS-LIT) web application suite for longitudinal MRI imaging follow-up of brain tumor patients. Specifically, this methodology combines the T1w-CE and FLAIR MRI data series with the original sMRI-based radiation treatment maps for each individual patient. Therefore, the purpose of the present study was to determine the recurrence timepoints for the patients enrolled in the dose-escalation clinical trial and perform median progression free survival (PFS) analysis based on the above methodology of interpreting imagistic data. Importantly, one limitation of the approach, which was fully acknowledged by the authors, is that the BrICS-LIT tools do not necessarily solve the problem of differentiating between tumor and necrosis on MRI data. 

While the authors state that their methodological approach could be used to overcome the confounding effects of pseudo-progression and radiation necrosis on the determination of true recurrence timepoints for these patients, it is not entirely clear if this can be truly achieved with the present methodology, especially when applied to in-field recurrences. The main weakness of the study is the overall small number of patient data sets (i.e., from 14 out of a total of 30 patients), with only 7 cases of in-field recurrences that ended up available for this type of analysis. Moreover, while a phase II clinical trial is certainly warranted by the generally promising outcomes of the pilot study, one potential downside of this dose escalation approach is the reliance on an increased number of biopsies and/or surgical reinterventions prompted by radiation-related events rather than true tumor recurrence events at the expense of additional risks and costs. Nonetheless, the authors acknowledge that a refinement of their methodology—i.e., by generating more defined Cho/NAA contours and radiation dose plans better tailored for each of these contours in order to further minimize patient toxicity and radiation-related events—will be needed in the future.  

My comments for the authors are as follows:

1.     Perhaps a better clarification regarding the potential limitations of the in-field recurrence determination based on imagistic data only following dose escalation and the need for a pathology assessment in these cases—although the frequency of these pathology assessments is another concern—will benefit the potential reader. 

2.     It would be interesting to know the MGMT promoter status (which could be reported in a table form) of the 14 patients whose data were analyzed with the present methodology. Specifically, it would be interesting to know whether there is any indication that some of the MGMT proficient tumors had benefited at all from the radiation dose escalation or not. This knowledge could be informative when designing a phase II clinical trial.   
